# Investigation of anxiety levels and associated factor analysis in breast cancer patients undergoing chemotherapy with implanted venous access ports during the stable phase of disease

Min Wen[1]☯, Yaling Zeng[2]☯, Purong Zhang 🔾[1]*, Jing Yang[1], Zhenjun Zhang[1], Baisen Li[1], Zhiyan Chen[1], Chunyan Fan[1], Qiao Zhang[1], Gui Yang[3], Li Wen[4]

1 Department of Breast, Sichuan Clinical Research Center for Cancer, Sichuan Cancer Hospital & Institute, Sichuan Cancer Center, Affiliated Cancer Hospital of University of Electronic Science and Technology of China, Chengdu, China, 2 School of Medicine, University of Electronic Science and Technology of China, Chengdu, Sichuan, China, 3 Department of Orthopedics, The Fourth People's Hospital of Nanchong City, Nanchong, Sichuan, China, 4 Sichuan Chengdu Narcotics Rehabilitation Center Hospital, Chengdu, Sichuan, China

☯ These authors contributed equally to this work.
* zprrong123321@163.com

## Abstract

### Aims

To analyze the occurrence of anxiety and its influencing factors in stable stage breast cancer patients after chemotherapy with Implanted Venous Access Port (IVAP).

### Methods

A total of 776 female breast cancer patients who underwent IVAP and completed chemotherapy in a tertiary cancer hospital from February 2019 to December 2021 were selected as the study subjects. General condition survey scale, sociological data, disease data, infusion port situation, and SAS anxiety scale were used for investigation to assess their anxiety status. Factors with statistically significant differences in univariate analysis were used as independent variables, and anxiety scale scores were used as dependent variables for multiple linear regression analysis of anxiety-related factors.

### Results

Among the 776 stable stage breast cancer patients after chemotherapy with IVAP, the average score of the anxiety scale was 50.35 ± 10.93, and the prevalence of anxiety was 58.51%, including 312 (40.21%) mild anxiety patients, 124 (15.98%) moderate anxiety patients, and 18 (2.32%) severe anxiety patients. The results of multivariate

**Data availability statement:** All relevant data are within the manuscript and its Supporting information files.

**Funding:** The author(s) received no specific funding for this work.

**Competing interests:** The authors have declared that no competing interests exist.

analysis showed that physical condition, wound dehiscence, pulling sensation, foreign body sensation, and the use of other venous channels and the order of chemotherapy and surgery were influencing factors of anxiety in stable stage breast cancer patients after implantable venous port chemotherapy (P < 0.05).

## Conclusions

Anxiety is common in stable stage breast cancer patients after chemotherapy with IVAP. Healthcare professionals need to pay attention to patients' psychological problems, strengthen psychological assessment, standardize and refine infusion port management, and promote patients' physical and psychological recovery.

## 1 Introduction

Breast cancer is the malignant tumor with the highest incidence in women in the world, and it shows an increasing trend annually [1]. According to the National Comprehensive Cancer Network (NCCN) Guidelines [2] (2024.V1), molecular subtypes of breast cancer (e.g., HR+/HER2-, triple-negative breast cancer) directly impact the selection of chemotherapy regimens and venous access routes. Particularly for patients requiring high-dose anthracyclines or taxanes, central venous access significantly reduces the risk of extravasation. As a standard treatment for breast cancer, chemotherapy has greatly enhanced patient survival rates [3]. Currently, the primary route for drug delivery in chemotherapy is intravenous infusion, typically administered through a central venous catheter in clinical practice to protect patients' blood vessels and reduce the discomfort associated with repeated venous punctures [4]. ESMO guidelines emphasize [5] that central venous access (including PICC, CVC, and implantable ports) is particularly indicated for administering vesicant/irritant agents (e.g., paclitaxel, carboplatin), thereby reducing risks of vascular damage and thrombotic complications. Catheterization sites primarily include subclavian vein puncture, internal jugular vein catheterization, peripherally inserted central venous catheters (PICC), and totally implantable venous access ports (IVAP). In 1988, the IVAP was first introduced in China. Liu's study [6] indicates a significantly lower overall infection risk with implantable venous access ports (IVAPs) versus peripherally inserted central catheters (PICCs) (OR: 0.570; 95% CI: 0.05–0.64), with clinically more significant advantages observed in breast cancer patients requiring long-term catheterization (>6 months). Compared to other venous access methods, its advantages include greater convenience and safety, longer dwell time, and the ability for repeated use [7,8]. As a result, IVAP is widely used for patients requiring long-term intermittent infusion and is also considered as an ideal access route for intravenous chemotherapy in cancer patients.

Stable breast cancer patients often choose to keep their IVAP after chemotherapy due to concerns about tumor recurrence. However, as patients reintegrate into their families and communities, they may experience long-term emotional distress due to concerns about changes in appearance, the need for regular maintenance, and

complications associated with the retention of the infusion port, potentially leading to a range of psychological issues [9]. Therefore, how to extend the survival period of stable breast cancer patients after IVAP chemotherapy while reducing their psychological issues has become a key focus for healthcare professionals. Based on this, this study intends to investigate the occurrence of anxiety in breast cancer patients undergoing implantable intravenous infusion chemotherapy during the stable stage of the disease, and analyze its related influencing factors, in order to provide a reference for early comprehensive interventions and more psychological attention for such patients.

## 2 Subjects and methods

### 2.1 Research subjects

The research subjects were female breast cancer patients who underwent implantable venous access surgery at Sichuan Cancer Hospital from September 1, 2019 to December 31, 2023 and completed chemotherapy as planned. This study has been reviewed by the Ethics Committee of the hospital (approval number: SCCEC-02-2024-112). After obtaining approval from the Institutional Review Board (IRB) of the affiliated hospital, the investigators completed the collection of data and scale assessments. This included the retrospective extraction of patients' clinicopathological data from the electronic medical record (EMR) system. Inclusion criteria:①Female patients diagnosed with breast cancer by pathological examination and completed intravenous chemotherapy by implantable intravenous infusion port, with clear follow-up results indicating no disease progression, and the disease is currently in a stable condition; ② Clear consciousness, and able to communicate normally and effectively;③Age ≥ 18 years old;④Informed consent and voluntary participation in this study. Exclusion criteria:①Combined with other malignant tumors;②People with severe mental disorders or cognitive impairment;③Suffering from other serious physical diseases.

### 2.2 Ethics statement

This study received initial approval from the Ethics Committee of Medical Research and New Medical Technology at Sichuan Cancer Hospital on May 30, 2024, followed by final ethical approval on June 26, 2024. This study enrolled 776 medically stable female breast cancer patients. These participants were individuals who had undergone breast cancer chemotherapy and implanted venous access port (IVAP) placement at our hospital between 2019 and 2023. The estimated sample size was 713, with a planned enrollment target of 700. Informed consent was typically obtained during routine electronic follow-up, outpatient IVAP maintenance visits, and scheduled outpatient disease surveillance appointments, as appropriate. Ultimately, 776 participants were enrolled in the study and provided written informed consent.

1) Ethics approval institution: Ethics Committee for Medical Research and New Medical Technology of Sichuan Cancer Hospital.

2) Ethics approval number: SCCEC-02-2024-112.

3) Ethics approval date: June 26, 2024.

4) Participants gave informed consent to participate in the study before taking part.

### 2.3 Research methods

**2.3.1 Study type and sample size calculation.** Employing a cross-sectional framework, this study conducted a retrospective analysis of cohort data. As reported in the existing literature [9], the prevalence rate of anxiety in breast cancer patients was 35.21%, which was set as 35% in this study. The sample size was estimated according to the formula $N = \frac{Ua^2P(1-P)}{d^2}$, p = 35%, allowable error d = 0.1, 95% confidence limit, α = 0.05, u(α/2)=1.96. This study utilized a consecutive sampling strategy. Based on the statistically estimated sample size, patients who received implanted venous access port

(IVAP) placement and completed chemotherapy at our Breast Center between January 1, 2019, and December 31, 2023, were enrolled. According to the inclusion and exclusion criteria of the study, a total of 920 eligible patients were screened out. Non-respondents comprised 68 cases (Deceased patients (N=4); Patients diagnosed with other malignancies, severe systemic diseases (N=5); Breast cancer recurrence and metastasis during follow-up (N=21); Patients unable to provide valid responses due to insufficient literacy (N=16); Refuse to answer (N=19); Suffering from severe mental disorders or cognitive impairments (N=3)). A total of 76 questionnaires were excluded as invalid due to the following reasons: Patients lost to follow-up (N=17); Patients opted for IVAP maintenance at other hospitals (N=34); Patients opted for IVAP disease surveillance at other hospitals (N=14); Patients with incomplete data entry (resulting in unavailable clinical catheterization information) (N=11), and a total of 776 respondents completed the survey, with a response rate of 84.3%, see Fig 1.

**2.3.2 General information questionnaire.** Ordinary circumstances, sociological data, disease data and infusion port information were collected through the general data questionnaire. Ordinary circumstances include age, education, employment, culture, place of residence, menopause, Body Mass Index (BMI), etc. Sociological data include employment status, monthly income per capita, marital status, payment method of medical expenses, ethnicity, etc. Disease data include diagnosis, disease stage, surgical method, chemotherapy regimen, post-chemotherapy period, current physical condition, underlying disease, etc. The conditions of infusion port include catheter length (cm), placement site, port placement time, X-ray positioning, local bleeding or bruising, selection of maintenance hospital at infusion port, maintenance cost, maintenance cycle, other venous access routes were used during the port placement period, etc.

In terms of "traction sensation" [10,11], it is defined as a persistent or activity-induced tightness, pulling or constricting sensation perceived by breast cancer patients during the indwelling of an implantable venous port (IVAP) at the catheter site or in the surrounding tissues, often accompanied by local discomfort or limited movement. This subjective feeling may be related to the mechanical stimulation of the catheter on the surrounding tissues, fibrotic response or increased neural sensitivity, and it is an important factor affecting the patient's psychological state (such as anxiety) and quality of daily life. The "foreign body sensation" [12,10] refers to the series of physiological and psychological reactions that an individual experiences when perceiving the existence of objects that do not belong to their normal tissues within or on their body

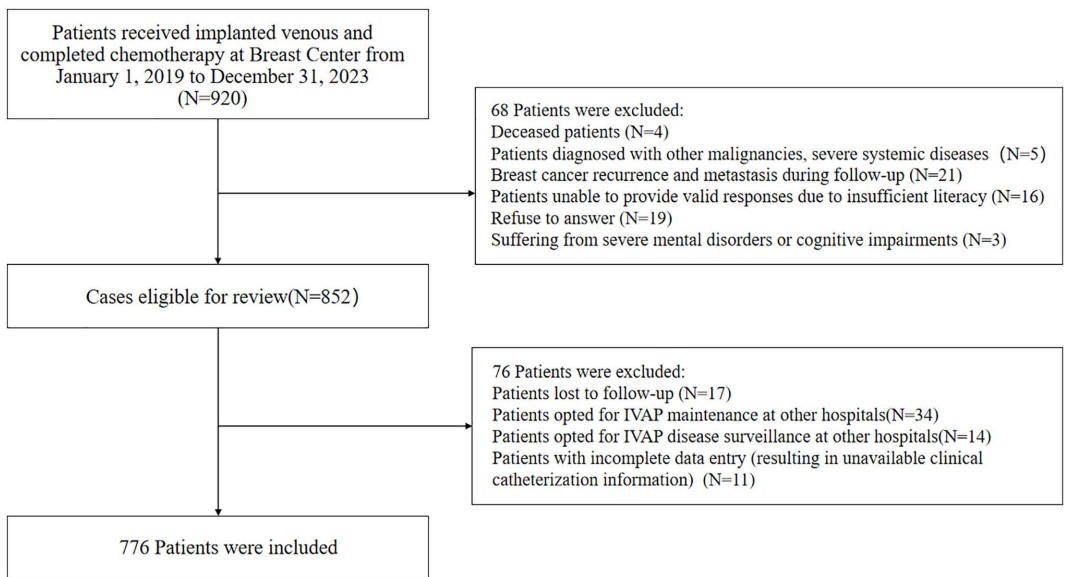

**Fig 1. Flowchart of Study Participant Selection.**

surface. In medical contexts (such as patients with PICC or IVAP catheterization), foreign body sensation not only involves the physical perception of the catheter's presence but also the emotional, cognitive and behavioral responses it triggers.

**2.3.3 Self-rating anxiety scale (SAS).** Compiled by William W.K. Zung et al. [13] in 1971, it is mainly used to analyze the subjective symptoms and feelings of patients, and is applicable to adults with anxiety symptoms. SAS is mainly assessed according to the frequency of symptoms defined for the project, which is divided into 4 levels, with 15 positive scores and 5 reverse scores. The score of 20 items is added to obtain a rough score, and the integer part is the standard score after multiplying the coefficient 1.25. SAS has a cut-off value of 50, 50–59 is mild anxiety, 60–69 is moderate anxiety, and 70 or more is severe anxiety.

The SAS scale has been widely verified in the generalized breast cancer population, and its reliability and validity are good (usually $\alpha > 0.80$). It has been widely used to assess the anxiety levels of patients with breast cancer and other malignant tumors. In recent domestic and international studies, SAS has been employed to investigate the psychological states of breast cancer patients before surgery, during the perioperative period, and during chemotherapy. The SAS score during the perioperative period can serve as an important indicator of intervention effectiveness [14]. Additionally, SAS has been utilized for breast cancer patients receiving chemotherapy through IVAP/PICC to evaluate the impact of different nursing models on anxiety [15]. Regarding the psychometric properties of the scale, the Cronbach's α of the original Zung SAS ranges from 0.81 to 0.92, and domestic large-sample studies on breast cancer patients report an α coefficient of approximately 0.88 [16], suggesting good consistency among the items. The SAS consists of 20 items, is concise and easy to understand, and is a self-rating scale, which can avoid social desirability bias caused by face-to-face interviews. It is suitable for rapid screening of anxiety status during outpatient follow-up or chemotherapy breaks. Therefore, SAS is applicable for the investigation and research of breast cancer patients undergoing chemotherapy through port-a-cath.

## 2.4 Questionnaire survey and quality control

The study involves the collection of data from patients undergoing maintenance of infusion ports, outpatient disease follow-up, and medication dispensing at vascular access clinics. A combination of electronic questionnaires and paper-based surveys will be utilized for this investigation.Before the investigation, the researchers provided a comprehensive explanation to the patients regarding the purpose and content of the study, as well as the time required for completing the questionnaire and ensuring confidentiality. Researchers usually informed consent was typically obtained during routine electronic follow-up, outpatient IVAP maintenance visits, and scheduled outpatient disease surveillance appointments, as appropriate. Upon obtaining informed consent from the patients, they signed an informed consent form.In the process, it is essential to utilize a consistent language and standardized guiding terminology; Respondents complete the questionnaire by themselves according to the actual situation; To ensure the qualification of the materials, all questionnaires will be collected and evaluated on-site. Questionnaires that cannot be remedied or are incomplete will be treated as invalid.

## 2.5 Statistical methods

The data analysis was conducted using SPSS 26.0 statistical software. The measurement data is described using $\bar{x} \pm s$, while comparisons of means between groups were performed using t-tests or analysis of variance (ANOVA). For multivariate analyses, multiple linear regression was utilized. The significance level was set at $\alpha = 0.05$, with $P < 0.05$ indicating statistically significant differences.

## 3 Research results

### 3.1 General information

This study included a total of 776 patients with stable-stage breast cancer who underwent chemotherapy via IVAP. The average age of the participants was $50.51 \pm 9.29$ years. Among them, 170 patients completed chemotherapy between 1–6

months prior, while 127 patients had finished their treatment between 6–12 months ago, and 479 patients had completed chemotherapy over one year ago. The average score on the anxiety scale was $50.35 \pm 10.93$, with an overall prevalence of anxiety at 58.51%. Among the participants, 312 individuals exhibited mild anxiety, corresponding to a prevalence rate of 40.21%. Additionally, there were 124 individuals classified as having moderate anxiety, representing a prevalence rate of 15.98%, while severe anxiety was identified in 18 individuals, accounting for a prevalence rate of 2.32%, see Fig 2.

### 3.2 Univariate analysis of anxiety levels in breast cancer patients during stable phase following IVAP chemotherapy

The 776 patients were classified according to different influencing factors: They were grouped according to demographic characteristics, which included age, BMI, ethnicity, education level, and the location of breast cancer diagnosis, etc, resulting in a total of 18 categories. They were divided into 17 categories according to port placement conditions, such as catheter length, insertion site, duration of placement, X-ray localization, and local bleeding or bruising, etc. It is divided into groups according to the maintenance status of the infusion ports, which includes hospitals that maintain infusion ports, associated maintenance costs, maintenance cycles for infusion ports, and other venous access methods.

**3.2.1  Impact of demographic factors on anxiety in patients with IVAP.**  The work situation, physical condition, underlying diseases, and type of chemotherapy are significant factors influencing anxiety in patients with port placement ($P < 0.05$). No statistically significant differences were observed in other demographic characteristics, as shown in Table 1.

**3.2.2  Impact of port-related factors on anxiety levels in patients with IVAP.**  Redness and swelling, poor wound healing, pulling sensation, swelling sensation, pain, foreign body sensation, itching or allergy, catheter blockage in the infusion port, and wound dehiscence are associated with the anxiety levels of patients with infusion ports ($P < 0.05$), as shown in Table 2.

**3.2.3  Impact of catheter maintenance practices on anxiety in patients with IVAP.**  During the period of port usage, the utilization of other venous access methods is associated with the anxiety levels of patients with infusion ports ($P < 0.05$), as shown in Table 3. Note: The reasons and proportions for the use of other venous access during the port placement period are as follows: There were 16 patients with emergency condition (10.39%), 92 patients (59.74%) faced a lack of conditions for using infusion ports at local hospitals, 22 patients (14.29%) with high cost, 12 patients (7.79%) expressed fear of pain associated with puncturing without damage needles, and 12 patients (7.79%) were unable to use their infusion ports due to complications that temporarily rendered them unusable.

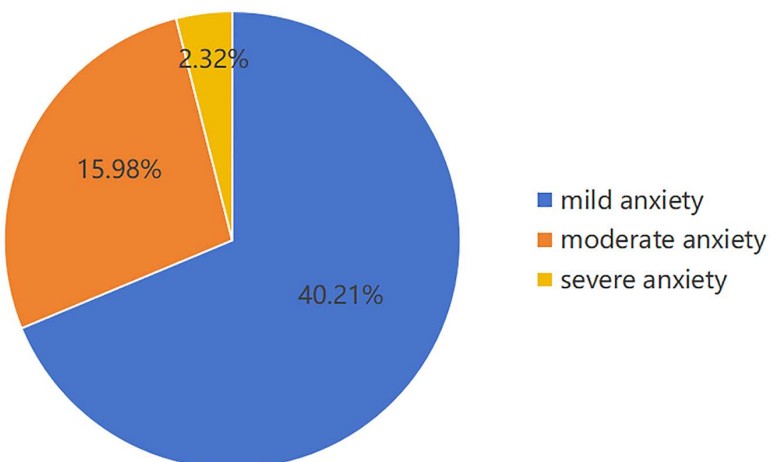

**Fig 2.  Distribution of Anxiety Severity in Patients.**

**Table 1. Demographic Characteristics and Their Association with Anxiety Scores (N = 776).**

| Variable | Category | n | Anxiety Score | t/F | P-value |
|---|---|---|---|---|---|
| Age (years) | <40 | 99 | 52.16±10.99 | 1.910 | 0.148 |
| | 40-60 | 588 | 50.14±10.88 | | |
| | >60 | 89 | 49.69±11.14 | | |
| BMI (kg/m²) | <18 | 13 | 54.77±14.02 | 1.638 | 0.195 |
| | 18-24 | 416 | 50.50±10.74 | | |
| | >24 | 347 | 49.99±11.02 | | |
| Nation | The Han nationality | 758 | 50.32±10.95 | 0.751 | 0.558 |
| | others | 18 | 51.39±10.42 | | |
| Educational level | Primary School and Below | 186 | 50.02±10.02 | 1.819 | 0.123 |
| | Junior high school | 243 | 49.73±11.27 | | |
| | High school/technical school/technical school | 166 | 50.58±11.44 | | |
| | College Diploma | 101 | 51.87±12.25 | | |
| | Bachelor degree or above | 80 | 50.56±8.95 | | |
| Diagnosis | Invasive carcinoma of left breast | 393 | 50.11±10.95 | 0.199 | 0.819 |
| | Invasive carcinoma of right breast | 369 | 50.58±10.70 | | |
| | Invasive carcinoma of both breasts | 14 | 51.00±16.21 | | |
| Clinical stage | Stage I | 168 | 50.97±11.51 | 0.499 | 0.737 |
| | Stage II | 406 | 50.39±10.30 | | |
| | Stage III | 134 | 49.25±10.15 | | |
| | Stage IV | 52 | 50.65±14.05 | | |
| | recrudescence | 16 | 50.94±15.19 | | |
| Chemotherapy regimen | TE/TEC/AC-T | 684 | 50.01±10.95 | 1.842 | 0.138 |
| | TCb | 39 | 52.54±12.38 | | |
| | EC/AC | 18 | 52.61±7.41 | | |
| | TC | 35 | 53.29±9.93 | | |
| Surgical Method | Total mastectomy | 571 | 49.97±11.09 | 1.528 | 0.218 |
| | Breast-conserving surgery | 185 | 51.57±10.51 | | |
| | Others | 20 | 49.90±9.79 | | |
| Working condition | unemployment | 394 | 50.47±11.71 | 2.964 | **0.031*** |
| | sick rest | 63 | 52.10±10.71 | | |
| | retirement | 177 | 48.45±9.78 | | |
| | employment | 142 | 51.59±9.84 | | |
| Per-capita household income | <2000 | 304 | 50.32±12.76 | 0.614 | 0.606 |
| | 2000—5000 | 341 | 49.96±9.80 | | |
| | 5000—10000 | 95 | 51.66±8.95 | | |
| | >10000 | 36 | 50.75±9.01 | | |
| Place of Residence | town | 243 | 51.12±12.55 | 1.783 | 0.182 |
| | village | 533 | 49.99±10.10 | | |
| Marital status | be married | 679 | 50.41±10.94 | 0.128 | 0.880 |
| | unmarried | 16 | 49.25±9.48 | | |
| | divorced/widowed | 81 | 50.02±11.22 | | |
| Medical insurance type | Employee medical insurance | 284 | 50.40±9.79 | 0.029 | 0.993 |
| | Medical insurance for urban residents | 183 | 50.37±11.54 | | |
| | The new rural cooperative medical insurance system | 274 | 49.83±9.44 | | |
| | others | 35 | 50.35±11.83 | | |

*(Continued)*

**Table 1.** (Continued)

| Variable | Category | n | Anxiety Score | t/F | P-value |
|---|---|---|---|---|---|
| Physical condition | good | 431 | 47.47 ± 10.40 | 41.119 | **<0.001*** |
| | discomfort | 309 | 53.41 ± 9.93 | | |
| | poor | 36 | 58.56 ± 13.91 | | |
| Underlying disease | have | 74 | 53.45 ± 10.99 | −2.573 | **0.010*** |
| | not | 702 | 50.02 ± 10.88 | | |
| Menopause | not | 86 | 49.91 ± 12.48 | −0.397 | 0.692 |
| | have | 690 | 50.40 ± 10.73 | | |
| Post-chemotherapy period | 1~6 months | 170 | 50.16 ± 11.90 | 0.079 | 0.924 |
| | 6~12 months | 127 | 50.67 ± 11.03 | | |
| | More than 1 year | 479 | 50.33 ± 10.56 | | |
| Surgery and chemotherapy implementation sequence | Surgery followed by chemotherapy | 448 | 51.10 ± 10.77 | 2.234 | **0.026*** |
| | Chemotherapy before surgery | 328 | 49.33 ± 11.09 | | |

Notes: 1. Significant results marked with**\*** (*P*<0.05). 2. SD: Standard Deviation; t: t-test; F: ANOVA. 3. Full table available in Supplementary Materials.

### 3.3 Multifactorial analysis of anxiety in breast cancer patients during the stable phase following IVAP chemotherapy

Prior to logistic regression analysis, tolerance (TOL) and variance inflation factor (VIF) were employed to assess multicollinearity among variables identified as significant in the univariate analysis. A TOL > 0.10 or VIF < 10.0 indicates the absence of significant multicollinearity. The results demonstrated TOL values ranging from 0.825 to 0.982 and VIF values ranging from 1.019 to 1.213 across all variables, confirming their suitability as predictor variables for inclusion in the logistic regression model. These findings indicate that none of the significant variables from the univariate analysis exhibited significant multicollinearity; therefore, all were included in the multivariate analysis, as shown in Table 4.

The factors identified as statistically significant in the univariate analysis were designated as independent variables, while the anxiety scale scores served as the dependent variable for multiple linear regression analysis. The variable assignment table is shown in Table 5.

The results of multiple factors showed that physical condition, wound dehiscence, pulling sensation, foreign body sensation, use of other venous access, as well as the surgery and chemotherapy implementation sequence are significant impact factors for anxiety in breast cancer patients during the stable phase following IVAP for chemotherapy (P < 0.05), as shown in Table 6.

## 4 Discussion

### 4.1 Elevated anxiety prevalence in breast cancer patients during post-chemotherapy stabilization with IVAP

Anxiety is a common emotional disorder among breast cancer patients, present during various stages of treatment and recovery. It not only exerts a negative impact on the psychological well-being of patients but also affects the efficacy of disease treatment, leading to a decline in patients' quality of life and potentially influencing survival rates [17]. In recent years, with the development of bio-psycho-social medical model, medical staff have increasingly recognized the importance of not only addressing patients' physiological ailments but also considering their psychological factors. In the studies on anxiety in breast cancer patients, scholars both domestically and internationally have primarily focused on patients during their treatment phases, such as those undergoing chemotherapy or post-operative care. There is a notable lack of research addressing the psychological health of patients in the home rehabilitation phase [18]. However, for breast cancer patients in a stable condition, particularly those who have achieved stability following IVAP chemotherapy, there is

**Table 2. Impact of different port placement conditions on patients' anxiety with IVAP.**

| Variable | Category | n | Anxiety Score | t/F | P-value |
|---|---|---|---|---|---|
| Catheter Length (cm) | <22 | 207 | 49.62 ± 11.28 | 1.203 | 0.301 |
| | 22-25 | 444 | 50.86 ± 10.60 | | |
| | >25 | 125 | 49.71 ± 11.47 | | |
| Placement site | Left neck in | 377 | 50.51 ± 11.09 | 0.154 | 0.694 |
| | Right neck in | 399 | 50.02 ± 10.78 | | |
| Port placement time | <40minutes | 510 | 49.87 ± 10.84 | −1.680 | 0.093 |
| | ≥40minutes | 266 | 51.26 ± 11.07 | | |
| X-ray positioning | 4-5 posterior ribs | 15 | 45.87 ± 11.801 | 1.284 | 0.275 |
| | 6 rear ribs | 90 | 50.79 ± 11.71 | | |
| | 7 Rear ribs | 396 | 50.39 ± 11.25 | | |
| | 8 rear ribs | 261 | 50.62 ± 10.13 | | |
| | 9 rear ribs | 14 | 45.93 ± 9.58 | | |
| Localized hemorrha-ge or ecchymosis | yes | 31 | 54.03 ± 11.95 | −1.919 | 0.055 |
| | no | 745 | 50.19 ± 10.87 | | |
| Infection | yes | 9 | 50.78 ± 10.93 | −0.119 | 0.906 |
| | no | 767 | 50.34 ± 12.04 | | |
| Red and swollen | yes | 32 | 56.84 ± 16.48 | −2.305 | **0.028*** |
| | no | 744 | 50.07 ± 10.55 | | |
| The wound is not healing well | yes | 23 | 54.91 ± 15.09 | −2.038 | **0.042*** |
| | no | 753 | 50.21 ± 10.76 | | |
| Pulling sensation | yes | 194 | 53.06 ± 11.02 | −4.033 | **<0.001*** |
| | no | 582 | 49.44 ± 10.76 | | |
| Sensation of swelli-ng | yes | 109 | 53.27 ± 12.92 | −2.607 | **0.010*** |
| | no | 667 | 49.87 ± 10.50 | | |
| Pain | yes | 100 | 53.76 ± 10.93 | −3.367 | **0.001*** |
| | no | 676 | 49.84 ± 10.85 | | |
| Foreign body sensa-tion | yes | 218 | 52.82 ± 10.72 | −3.977 | **<0.001*** |
| | no | 558 | 49.38 ± 11.09 | | |
| Itching or allergy | yes | 179 | 53.48 ± 10.61 | −4.424 | **<0.001*** |
| | no | 597 | 49.41 ± 10.86 | | |
| Thrombus | yes | 18 | 49.78 ± 9.11 | 0.224 | 0.823 |
| | no | 758 | 50.36 ± 10.97 | | |
| The infusion port is blocked | yes | 15 | 56.73 ± 12.79 | −2.291 | **0.022*** |
| | no | 761 | 50.22 ± 10.86 | | |
| Drug extravasation | yes | 2 | 52.50 ± 23.34 | −0.279 | 0.781 |
| | no | 774 | 50.34 ± 10.91 | | |
| Wound dehiscence | yes | 2 | 88.00 ± 24.04 | −4.951 | **<0.001*** |
| | no | 774 | 50.25 ± 10.74 | | |

Notes: 1. Significant results marked with* (P<0.05). 2. SD: Standard Deviation; t: t-test; F: ANOVA. 3. Full table available in Supplementary Materials.

currently a lack of detailed discussion in both domestic and international literature regarding the fluctuations in their anxiety levels and whether these warrant attention and early intervention.

This study included a total of 776 stable-stage breast cancer patients who had undergone chemotherapy with implanted venous infusion ports, with an average age of 50.51 ± 9.29 years old and an average score of 50.35 ± 10.93 on

the anxiety scale, among which the incidence of anxiety was 58.51%. This rate is higher than that reported by Hashemi et al. [16] (58.51% vs 41.9%). This discrepancy may be attributed to the fact that all participants in our study had experienced chemotherapy and implantation of venous infusion ports. On the one hand, the Guidelines and Standards for the Diagnosis and Treatment of Breast Cancer of the Chinese Anti-Cancer Association (2024 edition) recommend that

**Table 3. Impact of different port maintenance conditions on patients' anxiety with IVAP.**

| Variable | Category | n | Anxiety Score | t/F | P-value |
|---|---|---|---|---|---|
| Hospital for the mainte-nance of Infusion Ports | The hospital where they we-re treated | 516 | 50.56±10.97 | 0.975 | 0.404 |
| | Nearby or com-munity hospital | 72 | 49.74±10.54 | | |
| | Urban major Hospital | 46 | 52.13±10.42 | | |
| | Cancer hospital | 142 | 49.31±11.15 | | |
| Maintenance costs for infusion ports | <100 yuan | 61 | 52.20±10.75 | 2.330 | 0.055 |
| | 100-200 yuan | 268 | 48.76±10.10 | | |
| | 200-300 yuan | 211 | 51.06±10.90 | | |
| | 300~400 yuan | 94 | 50.87±10.48 | | |
| | ≥400 yuan | 142 | 51.14±12.53 | | |
| Maintenance cycle of infusion ports | 1 month | 240 | 50.55±10.22 | 0.673 | 0.569 |
| | 2 months | 276 | 50.76±11.37 | | |
| | 3 months | 207 | 50.00±10.59 | | |
| | ≥3 months | 53 | 48.62±12.92 | | |
| Use of other venous access | yes (1–2times) | 84 | 53.37±14.05 | 4.279 | **0.005*** |
| | no (3~5 times) | 27 | 53.96±12.15 | | |
| | yes (≥5 times) | 43 | 51.95±11.71 | | |
| | no | 622 | 49.67±10.24 | | |

Note: Reasons for the use of other venous access: ①the condition is urgent; ②the local hospital lacks the conditions for the use of infusion port; ③the cost is high; ④fear of pain associated with puncturing a non-damage needle; ⑤complications arising from the infusion port render it temporarily unusable.

**Table 4. Test for multicollinearity.**

| Factors | TOL | VIF |
|---|---|---|
| Working condition | 0.975 | 1.026 |
| Physical condition | 0.893 | 1.12 |
| Underlying disease | 0.939 | 1.065 |
| Surgery and chemotherapy implementation sequence | 0.982 | 1.019 |
| Redness and swelling | 0.825 | 1.213 |
| The wound is not healing well | 0.901 | 1.11 |
| Pulling sensation | 0.839 | 1.192 |
| Sensation of swelling | 0.902 | 1.109 |
| Pain | 0.852 | 1.174 |
| Foreign body sensation | 0.887 | 1.128 |
| Itching or allergy | 0.848 | 1.179 |
| The infusion port is blocked | 0.954 | 1.049 |
| Wound dehiscence | 0.85 | 1.177 |
| Use of other venous access | 0.943 | 1.06 |

patients with moderate and high risk of recurrence receive chemotherapy. This implies that these chemotherapy patients face a heightened risk of relapse, which may contribute to their anxiety. The underlying causes of this anxiety could stem from fears related to disease recurrence and metastasis, as well as physiological factors caused by toxic and side effects related to chemotherapy, which further exacerbate the psychological burden on patients [19]. On the other hand, in recipients of implantable venous access ports (IVAPs), the study by Janatolmakan et al. [20] demonstrated that prevalent deficiencies in knowledge of IVAP care protocols and aseptic management techniques contribute to diminished treatment self-efficacy, thereby predisposing patients to anxiety.

### 4.2 Predictors of anxiety in breast cancer patients with IVAP during post-chemotherapy stabilization

**4.2.1 Treatment-related physical symptomatology.** Grusdat et al. [21] found that after the diagnosis of breast cancer, women showed a decline in physical function and mental health, alongside pronounced symptoms of fatigue. Furthermore, there is a significant trend of deterioration following treatment. The findings of this study indicate that changes in quality of life and poor physical condition can impact patients' emotional states and mental health. In

**Table 5. Variable assignment table.**

| Variable | ID | Assignment specification |
|---|---|---|
| Anxiety score | Y | Raw value |
| Working condition | X1 | Unemployed = 1, sick leave = 2, retired = 3, employed = 4 |
| Physical condition | X2 | Good = 1, bad = 2, poor = 3 |
| Underlying disease | X3 | None = 0, With = 1 |
| Surgery and chemotherapy implementation sequence | X4 | Surgery followed by chemotherapy = 1, chemotherapy followed by surgery = 2 |
| The wound is not healing well<br>Redness and swelling | X5<br>X6 | None = 0, With = 1<br>None = 0, With = 1 |
| Pulling sensation | X7 | None = 0, With = 1 |
| Sensation of swelling | X8 | None = 0, With = 1 |
| Pain | X9 | None = 0, With = 1 |
| Foreign body sensation | X10 | None = 0, With = 1 |
| Itching or allergy | X11 | None = 0, With = 1 |
| The infusion port is blocked | X12 | None = 0, With = 1 |
| Wound dehiscence | X13 | None = 0, With = 1 |
| Use of other venous access | X14 | Yes (1–2 times) = 1, yes (3–5 times) = 2,<br>yes (more than 5 times) = 3, no = 4 |

**Table 6. Multivariate Linear Regression Analysis of Anxiety Predictors.**

| Predictor | β | SE | Standardized β | t-value | P-value |
|---|---|---|---|---|---|
| Physical condition | 5.091 | 0.630 | 0.273 | 8.075 | **<0.001*** |
| Wound dehiscence | 32.630 | 7.212 | 0.151 | 4.525 | **<0.001*** |
| Pulling sensation | 2.253 | 0.857 | 0.089 | 2.630 | **0.009*** |
| Foreign body sensation | 2.224 | 0.837 | 0.092 | 2.658 | **0.008*** |
| Use of other venous access | −1.067 | 0.377 | −0.096 | −2.834 | **0.005*** |
| Surgery and chemotherapy implementation sequence | −1.887 | 0.737 | −0.085 | −2.561 | **0.011*** |

this research, 4.62% of patients feel that their physical condition is poor, indicating that some patients may continue to experience significant physical discomfort after the completion of treatment. Furthermore, the severity of anxiety is correlated with a deterioration in overall health status, the more serious the anxiety, the worse the overall health status. This finding aligns with current relevant guidelines [22,23], which demonstrate that cancer-related fatigue (CRF) represents one of the most prevalent and persistent symptoms among breast cancer patients, frequently coexisting with significant psychological distress (e.g., anxiety, depression). These guidelines explicitly emphasize that both fatigue and emotional disturbances may persist throughout the entire treatment course and extended recovery phase.

Specifically regarding phenomena observed in this study, chemotherapy-induced adverse effects in breast cancer patients persist throughout the treatment continuum and recovery period. Beyond causing physical discomfort, these effects—particularly body image disturbances secondary to alopecia and illness experiences associated with long-term implantable port utilization—induce varied levels of psychological distress. In addition to causing physical discomfort, these side effects also lead to psychological trauma due to changes in appearance from hair loss and the ongoing experience of illness associated with long-term indwelling infusion ports. On the other hand, some patients with stable-stage breast cancer who have completed chemotherapy still experience poor physical condition and weakness, leading to a delay in their physical rehabilitation. In clinical practice, medical personnels need to further strengthen the management of physical symptoms and anxiety for these patients following chemotherapy with implanted venous access ports. This also underscores the importance of continuity in nursing care.

### 4.2.2 Port-site complications: Wound dehiscence, pulling sensation and foreign body sensation.

Although the advantages of intravenous infusion ports are prominent compared to traditional venous pathways, they can still lead to corresponding complications and discomfort. This may increase patients' physical unease and economic burden, ultimately resulting in adverse emotional responses. Domestic researchers focused on the occurrence and treatment of complications such as thrombus, infection, catheter blockage, catheter end displacement, catheter rupture, wound bleeding, and pinch-off syndrome in their studies on implantable intravenous infusion ports. In contrast, there has been relatively less attention paid to patients' feelings following catheter placement [24]. In this study, 25.0% of patients experienced a noticeable pulling sensation after the placement of the port, and 28.1% had foreign body sensation. It is noteworthy that, on the one hand, patients often experience anxiety and unease due to complications associated with infusion ports, fearing potential impacts on their physical health [25]. On the other hand, in the process of communication with patients, researchers in this study found that patients felt obvious pulling sensation and foreign body sensation after catheter insertion, which directly affected their sleeping posture, shoulder and neck activities, arm activities, etc., thereby affecting their daily lives and leading to adverse emotional responses among patients. In addition, the patients' insufficient understanding of venous access ports may lead to their concerns.

Therefore, for patients with implanted venous infusion ports, clinical attention should be directed towards the complications associated with the infusion port as well as the subjective feelings of the patients. Prior to port placement, medical personnels should not only inform patients about common complications related to the infusion port but also discuss potential changes in comfort levels that may arise from its use. Additionally, a standardized protocol for managing complications related to the infusion port should be established. Regular psychological assessments tailored to this patient group are essential, along with comprehensive health education initiatives. It is crucial to implement holistic management throughout the entire course of treatment for patients with infusion ports in order to alleviate anxiety stemming from complications and discomfort associated with their use.

### 4.2.3 Use of other venous access.

Among the 776 breast cancer patients in this survey, 622 (81.1%) patients who did not use other venous access had no anxiety (49.67 ± 10.24), and 154 (19.9%) patients had used other venous access due to other physical discomfort or anti-tumor treatment during hospitalization, indicating that this group of people had mild anxiety. Among them, 10.91% patients had used other venous access once or twice, and the score of anxiety scale was 53.37 ± 14.05. 3.47% of the patients had used other venous access 3~5 times, and the score of anxiety scale was

(53.96±12.15); 5.52% of the patients had used other venous access more than 5 times, and the anxiety scale score was (51.95±11.71).

The analysis of the reasons for using other venous access in this group of patients indicates that, in special scenarios, medical personnels must establish venous access rapidly. The use of an infusion port requires adherence to strict operational process, this procedure requires dedicated time for: strict aseptic technique throughout; pre-use port inspection/complication assessment; understanding port depth/thickness; selecting the correct non-coring (Huber) needle; aspirating blood and flushing after puncture; securing with a sterile transparent dressing shaped for close skin contact; and finally connecting to IV access. Compared to these, the infusion process using peripheral intravenous catheters and scalp veins is relatively straightforward, demonstrating a significant advantage in terms of time efficiency. The lack of access to infusion ports in local hospitals was the main reason for the use of other venous access, accounting for 59.74%, which may be related to the fact that 533 (68.7%) patients in this study were from rural areas. Due to the influence of geographical conditions and medical conditions, domestic township hospitals may not yet have implemented the techniques for port placement and maintenance of infusion ports, so these patients requiring intravenous infusion therapy are left with no choice but to opt for other venous access. In terms of costs, Fang et al. [8] indicated that the overall expenses associated with IVAP are higher than those of PICC. In this study, the total expenditure on catheter maintenance (including travel expenses and maintenance costs) indicates that only 7.86% of patients incurred total costs within 100 yuan, while 18.30% of patients had expenses amounting to 400 yuan or more. Furthermore, among the surveyed subjects, 39.18% of patients reported a monthly household income per capita below 2000 yuan, indicating that to a large extent, when patients experience other physical discomforts before the scheduled maintenance time for their infusion ports, they are more willing to choose other venous access to solve the current medical problems. In addition, patients at infusion port will experience more pain and discomfort during non-damaging needle punctures, which will not only increase patients' anxiety, but even lead them to evading therapeutic measures [26]. In this study, 7.79% of patients opted for other venous access due to for this reason. This further underscores the necessity for clinical medical staff to enhance their assessment of patients' pain and psychological state when dealing with infusion port patients. When appropriate, it is essential to implement safe and effective methods that can alleviate pain associated with non-invasive needle punctures, thereby improving patient comfort.

**4.2.4 Surgery and chemotherapy implementation sequence.** The systemic chemotherapy for early-stage breast cancer is categorized into neoadjuvant chemotherapy and adjuvant chemotherapy following surgery. This study indicates that among 766 patients, 57.73% received postoperative adjuvant chemotherapy, while 42.27% underwent neoadjuvant chemotherapy. The results reveal that patients who received postoperative adjuvant chemotherapy exhibited more severe anxiety symptoms, which is consistent with findings from scholars both domestically and internationally [27]. The study of Yang et al. [28] indicates that postoperative patients must endure the pain caused by various treatments, as well as the fear associated with their illness, which was the most serious period of patients' anxiety. On one hand, the impairment of body image resulting from surgery is likely a key factor influencing patients' emotional well-being. On the other hand, following surgery, patients often experience a decline in overall bodily function. Coupled with adverse physical conditions and the toxic side effects of chemotherapy drugs,such as gastrointestinal reactions, hair loss, and insomnia, etc., these pronounced somatic symptoms make patients more susceptible to feelings of anxiety.

Therefore, for patients with postoperative adjuvant chemotherapy, clinical interventions can be implemented to assist breast cancer patients in alleviating discomfort caused by surgery and reducing the occurrence of chemotherapy side effects, thereby promoting their physical recovery. At the same time, attention should be paid to the psychological care of patients undergoing postoperative chemotherapy, providing targeted nursing interventions helps patients to accurately understand their condition, actively cope with the negative emotions associated with surgery and chemotherapy, and alleviate anxiety-related psychological disorders [29].

## 5 Conclusion

In summary, the independent factors influencing anxiety in patients with stable-stage breast cancer following chemotherapy via implanted venous infusion ports include physical condition, wound dehiscence, pulling sensation, foreign body sensation, use of other venous access, and the sequence of chemotherapy and surgery. The research findings indicate that patients with stable breast cancer following IVAP chemotherapy exhibit elevated levels of anxiety, and the situation is not optimistic. In clinical practice, it is essential to enhance awareness of the use of infusion ports and to implement health education based on postoperative rehabilitation. Particularly for these patients, it is crucial to provide increased social psychological services and humanistic care. The focus of emotional management of breast cancer should be focused on patients' physical condition, discomfort and complications brought about by port placement, as well as whether their living environments are suitable for using infusion ports and postoperative adjuvant chemotherapy patients. And the measures that are beneficial to improving the above factors should be the priority direction of managing breast cancer anxiety.

## 6 Limitations

This study has several limitations:

a) Limited generalizability: Single-center recruitment with restricted sample size may introduce homogeneity in demographics, disease profiles, and institutional practices, constraining external validity.

b) Measurement biases: Self-reported anxiety (via SAS) risks response/recall biases; Lack of structured clinical interviews (e.g., SCID) prevents diagnostic differentiation; Unvalidated SAS psychometrics in port-using chemotherapy cohorts

c) Uncontrolled confounders: Inadequate accounting for psychosocial interventions and pre-existing mental health conditions.

d) Temporal constraints: Short post-chemotherapy assessment window precludes evaluation of long-term psychological effects.

## 7 Future perspectives

For future studies, we will conduct multicenter collaborations to recruit a larger, geographically diverse patient population, enhancing the representativeness and generalizability of our findings. We will implement a hybrid assessment approach combining semi-standardized self-report scales with structured clinical interviews based on diagnostic criteria, thereby improving the accuracy and diagnostic precision of anxiety evaluations. Where feasible, prospective multipoint assessments will be employed to minimize recall bias and delineate anxiety trajectories in greater detail. Psychosocial support interventions and comprehensive mental health histories (including diagnoses and treatments) will be systematically documented and analyzed. Statistical adjustments will be applied as needed to clarify relationships between core variables.

## Supporting information

**S1 File. Supplementary data files.** https://doi.org/10.6084/m9.figshare.29959559.
(XLSX)

## Author contributions

**Conceptualization:** Min Wen, Yaling Zeng, Purong Zhang.

**Data curation:** Jing Yang, Zhenjun Zhang, Baisen Li, Zhiyan Chen, Chunyan Fan, Qiao Zhang.

**Formal analysis:** Min Wen, Yaling Zeng, Jing Yang, Zhenjun Zhang, Baisen Li, Gui Yang, Li Wen.

**Funding acquisition:** Purong Zhang.

**Investigation:** Jing Yang, Zhenjun Zhang, Baisen Li, Gui Yang, Li Wen.

**Methodology:** Min Wen, Yaling Zeng.

**Project administration:** Purong Zhang.

**Supervision:** Purong Zhang.

**Validation:** Zhiyan Chen, Chunyan Fan, Qiao Zhang.

**Writing – original draft:** Min Wen, Yaling Zeng.

**Writing – review & editing:** Purong Zhang.

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
