## [Decision Letter · Decision Letter 0]

1 Jul 2025

PONE-D-25-12070Investigation of Anxiety Levels and Associated Factor Analysis in Breast Cancer Patients Undergoing Chemotherapy with Implanted Venous Access Ports During the Stable Phase of DiseasePLOS ONE

Dear Dr. Zhang,

Thank you for submitting your manuscript to PLOS ONE. After careful consideration, we feel that it has merit but does not fully meet PLOS ONE’s publication criteria as it currently stands. Therefore, we invite you to submit a revised version of the manuscript that addresses the points raised during the review process.

The manuscript addresses a clinically relevant issue concerning anxiety among breast cancer survivors with Implanted Venous Access Ports (IVAPs), and falls within the scope of PLOS ONE. However, major revisions are required before it can be considered for publication. Key issues include a lack of clarity regarding the timeline of ethics approval versus data collection, absence of psychometric validation or internal consistency reporting for the anxiety measure used, and insufficient adjustment for confounding variables in the regression analysis. The authors must also clarify their sampling strategy, handling of missing data, and ensure that all references are complete and correctly formatted. The manuscript requires significant English language editing to improve clarity. Although not essential, I strongly recommend expanding the literature review to include international psycho-oncology guidelines and revising the tables for readability. Both reviewers raised consistent and valid concerns, which the authors should address comprehensively

We look forward to receiving your revised manuscript.

Kind regards,

Sampath Kumar Amaravadi, Ph.D

Academic Editor

PLOS ONE

Journal Requirements:

For additional information about PLOS ONE ethical requirements for human subjects research, please refer to http://journals.plos.org/plosone/s/submission-guidelines#loc-human-subjects-research .

Additional Editor Comments:

Recommended Revisions 

Literature Review

The introduction and discussion would benefit from greater engagement with international literature (e.g., NCCN, ESMO, psycho-oncology guidelines) to situate the findings in a broader global context, particularly for a PLOS ONE readership.

Tables and Visual Aids

Consider revising the tables for readability—highlight significant results, clarify column headings, and use consistent formatting. Including a participant flow diagram and a figure showing anxiety severity distribution would improve the manuscript’s accessibility.

Operational Definitions

Brief operational definitions of key subjective terms (e.g., “pulling sensation”) should be included to ensure interpretability across clinical contexts.

Reviewers' comments:

Reviewer's Responses to Questions

**Comments to the Author**

1. Is the manuscript technically sound, and do the data support the conclusions?

Reviewer #1: Yes

Reviewer #2: Yes

2. Has the statistical analysis been performed appropriately and rigorously? 

Reviewer #1: Yes

Reviewer #2: Yes

3. Have the authors made all data underlying the findings in their manuscript fully available?

Reviewer #1: Yes

Reviewer #2: Yes

4. Is the manuscript presented in an intelligible fashion and written in standard English?

Reviewer #1: Yes

Reviewer #2: Yes

5. Review Comments to the Author

Reviewer #1: • Sampling technique not well-detailed.

Suggestion: Kindly clarify the sampling approach used to recruit the 920 eligible participants. For example, indicate whether a consecutive or convenience sampling strategy was employed. This will help readers evaluate the representativeness of the study population and potential biases.

• No discussion of non-responders or missing data.

Suggestion: Please include a brief discussion on non-responders or invalid questionnaires. Were there any systematic differences (e.g., demographic or clinical characteristics) between those who responded and those who did not? Even a statement acknowledging the limitation of not assessing this would be helpful.

• No psychometric validation of SAS in this specific cohort (e.g., internal consistency): Even though the SAS is a validated tool generally, its reliability can vary across populations, especially when an individual have a life threatening conditions like cancer. Without testing internal consistency in this population, it’s unclear whether the scale is truly measuring anxiety accurately and consistently in these patients.

Suggestion: Please consider reporting the internal consistency of the SAS scale in your sample (e.g., Cronbach’s alpha). Although SAS is widely validated, reporting this psychometric property will improve the reliability and generalizability of your anxiety findings in this unique cohort of breast cancer patients post-IVAP chemotherapy. If already available, this can be easily added to the Methods and Results sections. Or this could be mentioned in the limitation.

• Improve Reporting of Variable Definitions.

Suggestion: It would enhance the clarity and reproducibility of your study if you provide brief operational definitions or criteria used to assess subjective symptoms like “pulling sensation” or “foreign body sensation.” This will help standardize interpretation across different clinical contexts.

Reviewer #2: Overall Summary of the Research

This study investigates the prevalence and associated factors of anxiety among female breast cancer patients in a stable disease phase who previously received chemotherapy through Implanted Venous Access Ports (IVAPs). Conducted at a single tertiary cancer hospital in China, the study involved 776 patients, using self-administered questionnaires and the Zung Self-Rating Anxiety Scale (SAS). Through univariate and multivariate regression analysis, the study identified several factors significantly associated with anxiety levels, including physical condition, wound dehiscence, sensations related to the IVAP (e.g., pulling or foreign body sensation), use of alternative venous access, and the sequence of surgery and chemotherapy.

The manuscript addresses an important yet underexplored dimension of cancer survivorship care. While the research question is relevant and the sample size substantial, the study has some limitations regarding methodological structure, literature review, and manuscript structure that must be addressed prior to publication.

MAJOR ISSUES

1. Study Design and Clarity- There is confusion about the timeline of the study. The ethics approval is dated June 2024, while data were collected from patients treated between 2019 and 2023. It is unclear how retrospective consent was obtained. The methods need clear alignment of ethics, data access, and recruitment periods. The study is described as cross-sectional, but it includes retrospective data analysis. The authors should explicitly state this and address related limitations (e.g., recall bias).

2. Instrument Validation- There is no report of cultural or linguistic validation of the Zung SAS for use in this specific patient population. Please clarify whether the version used has been validated in Chinese (if applicable), and provide a reference.

3. Confounding and Covariates- The regression model does not appear to adjust for potential confounders such as age, cancer stage, treatment type, or socioeconomic status, which could influence anxiety. These should be included or justified for exclusion. It is unclear how multicollinearity was checked in the regression model, especially given several closely related variables (e.g., symptoms from IVAP, physical condition, etc.).

4. Literature Review and Theoretical Framing- The introduction and discussion underutilise international literature. For a PLOS ONE audience, broader referencing beyond national studies (e.g., NCCN or ESMO psycho-oncology guidelines) would strengthen the scientific context. Some references are missing or shown as placeholders (e.g., “Error! Reference source not found.”). Please revise for completeness and accuracy.

5. Language and Grammar- The manuscript requires extensive English language editing to improve clarity and fluency. Grammatical errors, awkward phrasing, and overly literal translations (e.g., "bad physical condition") detract from the overall quality.

MINOR ISSUES

1. Tables and Figures- Tables are dense and could benefit from clearer headings, footnotes, and consistent formatting. Consider highlighting statistically significant results for easier interpretation. A flow diagram of patient inclusion/exclusion and a figure visualising anxiety severity distribution would enhance clarity.

2. Ethical Clarity- The ethics section mentions both “preliminary review” and “final ethics approval” dated after data collection began. Please clarify when ethical approval was granted in relation to participant recruitment.

3. Limitations- The limitations section is quite brief. It should discuss:

a) Single-centre recruitment

b) Use of self-reported outcomes

c) Lack of clinical anxiety diagnosis (vs. screening scale)

d) No data on supportive care interventions or prior psychiatric history

4. Data Availability- While the manuscript states that all data are available in supplementary files, ensure that datasets are in a machine-readable format and include a DOI if applicable, in line with PLOS data policies.

6. PLOS authors have the option to publish the peer review history of their article (what does this mean? ). If published, this will include your full peer review and any attached files.

**Do you want your identity to be public for this peer review?** For information about this choice, including consent withdrawal, please see our Privacy Policy .

Reviewer #1: No

Reviewer #2: **Yes: ** Khyati Manoj Shah

---

## [Author Response · Author response to Decision Letter 1]

24 Aug 2025

Response

Reviewer 1

• Sampling technique not well-detailed.

Suggestion: Kindly clarify the sampling approach used to recruit the 920 eligible participants. For example, indicate whether a consecutive or convenience sampling strategy was employed. This will help readers evaluate the representativeness of the study population and potential biases.

Responses:

Thank you for the reviewers' suggestions. This study adopted a continuous sampling strategy. Based on the estimated sample size from the research statistics, patients who received implanted venous access ports and completed chemotherapy at the breast center of our hospital from January 1, 2019 to December 31, 2023 were included. According to the inclusion and exclusion criteria of the study, a total of 920 eligible patients were screened out. Non-respondents comprised 68 cases (Deceased patients (N=4); Patients diagnosed with other malignancies, severe systemic diseases (N=5); Breast cancer recurrence and metastasis during follow-up (N=21); Patients unable to provide valid responses due to insufficient literacy (N=16); Refuse to answer (N=19); Suffering from severe mental disorders or cognitive impairments (N=3)). A total of 76 questionnaires were excluded as invalid due to the following reasons: Patients lost to follow-up (N=17); Patients opted for IVAP maintenance at other hospitals (N=34); Patients opted for IVAP disease surveillance at other hospitals (N=14); Patients with incomplete data entry (resulting in unavailable clinical catheterization information) (N=11). Finally, 776 cases were included in the analysis questionnaire, with an effective response rate of 84.3%. This has been elaborated in the method section of the article, and the relevant tables of the patient screening flowchart have been supplemented in the research method section.

Figure 1. Flowchart of Study Participant Selection

• No discussion of non-responders or missing data.

Suggestion: Please include a brief discussion on non-responders or invalid questionnaires. Were there any systematic differences (e.g., demographic or clinical characteristics) between those who responded and those who did not? Even a statement acknowledging the limitation of not assessing this would be helpful.

Responses:

Thank you for the reviewers' suggestions. In this study, a total of 920 questionnaires were distributed. Of these, 776 valid questionnaires were retrieved. Non-respondents comprised 68 cases (Deceased patients (N=4); Patients diagnosed with other malignancies, severe systemic diseases (N=5); Breast cancer recurrence and metastasis during follow-up (N=21); Patients unable to provide valid responses due to insufficient literacy (N=16); Refuse to answer (N=19); Suffering from severe mental disorders or cognitive impairments (N=3)). A total of 76 questionnaires were excluded as invalid due to the following reasons: Patients lost to follow-up (N=17); Patients opted for IVAP maintenance at other hospitals (N=34); Patients opted for IVAP disease surveillance at other hospitals (N=14); Patients with incomplete data entry (resulting in unavailable clinical catheterization information) (N=11). Finally, a total of 144 cases were not included in the analysis.

During the statistical process, the researchers attempted to compare the demographic and clinical characteristics between the responders and the non-responders. Through the independent samples t-test and one-way analysis of variance (ANOVA) revealed no statistically significant differences, with all P�0.05. However, among these 144 patients, the number of non-responders has limited access to relevant information, and there is still a possibility that the differences have not been identified. Therefore, a total of 776 valid questionnaires were obtained in the end, with an effective response rate of 84.3%.

Although the research team made every effort to verify the differences, due to the limitations of actual conditions, it was impossible to completely rule out the potential differences that might exist. This is indeed the limitation of sample inclusion in this study. Supplementary discussions have been made in the section on the limitations of the manuscript.

• No psychometric validation of SAS in this specific cohort (e.g., internal consistency): Even though the SAS is a validated tool generally, its reliability can vary across populations, especially when an individual have a life threatening conditions like cancer. Without testing internal consistency in this population, it’s unclear whether the scale is truly measuring anxiety accurately and consistently in these patients.

Suggestion: Please consider reporting the internal consistency of the SAS scale in your sample (e.g., Cronbach’s alpha). Although SAS is widely validated, reporting this psychometric property will improve the reliability and generalizability of your anxiety findings in this unique cohort of breast cancer patients post-IVAP chemotherapy. If already available, this can be easily added to the Methods and Results sections. Or this could be mentioned in the limitation.

Responses:

1.Thank you to the reviewers for your valuable comments We have conducted a comprehensive literature search and have not yet found any reports on the internal consistency of the SAS Self-rating Anxiety Scale (such as Cronbach's α) in the population of breast cancer patients undergoing port of infusion (IVAP) chemotherapy. However, the SAS scale has been widely verified in the generalized breast cancer population, and its reliability and validity are good (usually α>0.80). In view of this situation, we have supplemented this issue in the the methods and results sections.

2.The Self-Rating Anxiety Scale (SAS) has been widely used to assess the anxiety levels of patients with breast cancer and other malignant tumors. In recent domestic and international studies, SAS has been employed to investigate the psychological states of breast cancer patients before surgery, during the perioperative period, and during chemotherapy. The SAS score during the perioperative period can serve as an important indicator of intervention effectiveness [1]. Additionally, SAS has been utilized for breast cancer patients receiving chemotherapy through IVAP/PICC to evaluate the impact of different nursing models on anxiety [2]. Regarding the psychometric properties of the scale, the Cronbach's α of the original Zung SAS ranges from 0.81 to 0.92, and domestic large-sample studies on breast cancer patients report an α coefficient of approximately 0.88 [3], suggesting good consistency among the items. The SAS consists of 20 items, is concise and easy to understand, and is a self-rating scale, which can avoid social desirability bias caused by face-to-face interviews. It is suitable for rapid screening of anxiety status during outpatient follow-up or chemotherapy breaks. Therefore, SAS is applicable for the investigation and research of breast cancer patients undergoing chemotherapy through port-a-cath.

References:

[1]Li J, Gao W, Yang Q, Cao F. Perceived stress, anxiety, and depression in treatment-naïve women with breast cancer: a case-control study. Psychooncology. 2021;30(2):231-239. doi:10.1002/pon.5555.

[2]Kim YH, Choi KS, Han K, Kim HW. A psychological intervention programme for patients with breast cancer under chemotherapy and at a high risk of depression: A randomised clinical trial. J Clin Nurs. 2018;27(3-4):572-581. doi:10.1111/jocn.13910.

[3]Hashemi SM, Rafiemanesh H, Aghamohammadi T, et al. Prevalence of anxiety among breast cancer patients: a systematic review and meta-analysis. Breast Cancer. 2020;27(2):166-178. doi:10.1007/s12282-019-01031-9.

• Improve Reporting of Variable Definitions.

Suggestion: It would enhance the clarity and reproducibility of your study if you provide brief operational definitions or criteria used to assess subjective symptoms like “pulling sensation” or “foreign body sensation.” This will help standardize interpretation across different clinical contexts.

Responses:

Thank you for your valuable suggestions. We fully recognize that clear variable definitions are crucial for the clarity and reproducibility of research. This has been detailed in the Methodology section.

For subjective symptoms such as "pulling sensation" and "foreign body sensation", the research team, by referring to existing literature and taking into account the actual situation of clinical patients, has made a detailed summary and elaboration of their definitions in the research methods section of the manuscript. In terms of "traction sensation", it is defined as a persistent or activity-induced tightness, pulling or constricting sensation perceived by breast cancer patients during the indwelling of an implantable venous port (IVAP) at the catheter site or in the surrounding tissues, often accompanied by local discomfort or limited movement. This subjective feeling may be related to the mechanical stimulation of the catheter on the surrounding tissues, fibrotic response or increased neural sensitivity, and it is an important factor affecting the patient's psychological state (such as anxiety) and quality of daily life. The "foreign body sensation" refers to the series of physiological and psychological reactions that an individual experiences when perceiving the existence of objects that do not belong to their normal tissues within or on their body surface. In medical contexts (such as patients with PICC or IVAP catheterization), foreign body sensation not only involves the physical perception of the catheter's presence but also the emotional, cognitive and behavioral responses it triggers.

We believe that these definitions are helpful in standardizing the interpretation of symptoms in different clinical contexts and improving the overall quality of research. Thank you again for your guidance.

References

[1]Voog E, Campion L, du Rusquec P, et al. Totally implantable venous access ports: a prospective long-term study of early and late complications in adult patients with cancer. Support Care Cancer. 2018;26(1):81-89. doi:10.1007/s00520-017-3816-3.

[2]Ignatov A, Hoffman O, Smith B, et al. An 11-year retrospective study of totally implanted central venous access ports: complications and patient satisfaction. Eur J Surg Oncol. 2009;35(3):241-246. doi:10.1016/j.ejso.2008.01.020.

[3]Schiffer CA, Mangu PB, Wade JC, et al. Central venous catheter care for the patient with cancer: American Society of Clinical Oncology clinical practice guideline. J Clin Oncol. 2013;31(10):1357-1370. doi:10.1200/JCO.2012.45.5733.

[4]Voog E, Campion L, du Rusquec P, et al. Totally implantable venous access ports: a prospective long-term study of early and late complications in adult patients with cancer. Support Care Cancer. 2018;26(1):81-89. doi:10.1007/s00520-017-3816-3.

Reviewer 2

• Overall Summary of the Research

This study investigates the prevalence and associated factors of anxiety among female breast cancer patients in a stable disease phase who previously received chemotherapy through Implanted Venous Access Ports (IVAPs). Conducted at a single tertiary cancer hospital in China, the study involved 776 patients, using self-administered questionnaires and the Zung Self-Rating Anxiety Scale (SAS). Through univariate and multivariate regression analysis, the study identified several factors significantly associated with anxiety levels, including physical condition, wound dehiscence, sensations related to the IVAP (e.g., pulling or foreign body sensation), use of alternative venous access, and the sequence of surgery and chemotherapy.The manuscript addresses an important yet underexplored dimension of cancer survivorship care. While the research question is relevant and the sample size substantial, the study has some limitations regarding methodological structure, literature review, and manuscript structure that must be addressed prior to publication.

Responses:

Sincerely thank the reviewers for their comments! We have revised and improved the structure of the methods we are studying, the literature review and the manuscript structure. You can view them in the revised manuscript section. Thank you sincerely again

MAJOR ISSUES

• Study Design and Clarity- There is confusion about the timeline of the study.

The ethics approval is dated June 2024, while data were collected from patients treated between 2019 and 2023. It is unclear how retrospective consent was obtained. The methods need clear alignment of ethics, data access, and recruitment periods. The study is described as cross-sectional, but it includes retrospective data analysis. The authors should explicitly state this and address related limitations (e.g., recall bias).

Responses:

Dear reviewers, thank you very much for your review of our research and your valuable suggestions. In response to your questions regarding the research timeline and methods, we hereby provide further explanations and clarifications.

Firstly, regarding the acquisition of retrospective consent: At the authors' affiliated hospital, ethical review requires successful completion of a preliminary assessment prior to granting full certification. The study protocol received preliminary IRB clearance on 30 May 2024 , with formal certification issued on 26 June 2024 (IRB Approval No: SCCHEC-02-2024-112). Participant recruitment was initiated strictly after obtaining the final IRB certification document. The research subjects were patients who had received chemotherapy for breast cancer and received IVAP implantation at the research team hospital from 2019 to 2023. The data collection was divided into two parts: one part was the collection of questionnaires, and the other part was the collection of relevant retrospective clinical medical records of the patients through the hospital's electronic information system and outpatient intravenous care system. Retroactive consent was obtained in written or oral form, and we have added this statement in the revised draft. The hospital where the research team is located is the largest specialized cancer hospital in the southwest of China, with a wide range of patient sources. Member Wen Min is a full-time follow-up management staff member. She regularly follows up with patients through phone calls, wechat, etc. Generally, patients will complete informed consent during regular electronic follow-ups, maintenance of outpatient intravenous access (IVAP), and outpatient disease re-examination cycles.

Secondly, regarding the research design: We confirm that this study adopted a cross-sectional design and included the part of retrospective data analysis. We will clearly elaborate on this design feature in the revised draft. Given the retrospective nature of the research data (i.e., reliance on participants' recollection of past exposure or outcome events), recall bias is an inherent methodological limitation of this design that is difficult to completely avoid. When discussing the limitations of the research, we will focus on elaborating on this, exploring its potential impact on the research results (such as the intensity or direction of exposure-outcome associations), as well as the efforts we have made during the research process to minimize its impact (for example: using validated structured questionnaires, controlling the interview environment, setting clear recall time anchors, etc.).

Finally, we would like to express our sincere gratitude once again for your review and guidance. We will carefully consider your suggestions and make corresponding modifications in the revised draft. We look forward to your valuable feedback again.

• Instrument Validation- There is no report of cultural or linguistic validation of the Zung SAS for use in this specific patient population. Please clarify whether the version used has been validated in Chinese (if applicable), and provide a reference.

Responses:

We thank the reviewers for their insightful comments. The Zung Self-Rating Anxiety Scale (SAS) used in this study is the officially validated Chinese version, which has undergone cultural and linguistic adaptation. Its reliability and validity have been established across multiple populations in China. The relevant validation studies and sources are outli

---

## [Editor Report · Decision Letter 1]

5 Sep 2025

Investigation of Anxiety Levels and Associated Factor Analysis in Breast Cancer Patients Undergoing Chemotherapy with Implanted Venous Access Ports During the Stable Phase of Disease

PONE-D-25-12070R1

Dear Dr. Zhang,

We’re pleased to inform you that your manuscript has been judged scientifically suitable for publication and will be formally accepted for publication once it meets all outstanding technical requirements.

Kind regards,

Sampath Kumar Amaravadi, Ph.D

Academic Editor

PLOS ONE

---

## [Editor Report · Acceptance letter]

PONE-D-25-12070R1

PLOS ONE

Dear Dr. Zhang,

I'm pleased to inform you that your manuscript has been deemed suitable for publication in PLOS ONE. Congratulations! Your manuscript is now being handed over to our production team.

Kind regards,

on behalf of

Dr Sampath Kumar Amaravadi

Academic Editor

PLOS ONE